# Public Perceptions of Harms and Benefit of COVID-19 Immunity Certificate: A Cross-Sectional Study in the Italian Setting

**DOI:** 10.3390/vaccines10091501

**Published:** 2022-09-08

**Authors:** Serena Barello, Michele Paleologo, Lorenzo Palamenghi, Marta Acampora, Guendalina Graffigna

**Affiliations:** 1Faculty of Psychology, Università Cattolica del Sacro Cuore, L.go Gemelli 1, 20123 Milan, Italy; 2Department of Psychology, Università Cattolica del Sacro Cuore, L.go Gemelli 1, 20123 Milan, Italy; 3EngageMinds HUB—Consumer, Food & Health Engagement Research Center, Università Cattolica del Sacro Cuore, 20123 Milan, Italy; 4EngageMinds HUB—Consumer, Food & Health Engagement Research Center, 26100 Cremona, Italy; 5Faculty of Agriculture, Food and Environmental Sciences, Università Cattolica del Sacro Cuore, Via Bissolati, 74, 26100 Cremona, Italy

**Keywords:** health certificate, immunity passports, immunity certification, public responses, health policy, consumer psychology

## Abstract

A cross-sectional survey between 29 January 2022 and 3 February 2022 was conducted to understand the public rationale for accepting or rejecting the use of COVID-19 immunity certificates and to identify the psychosocial factors that mostly predict the positive/negative individuals’ perceptions of this measure. One thousand twenty-two Italian adults were recruited by a professional panel provider by employing a stratified sampling strategy controlled for gender, age, geographical area of residence, size of the urban centre of residence, employment, and wage. Eight Welch’s ANOVAs were then carried out to compare the perception of benefits and the perception of harms among different population groups. Multiple linear regression was carried out to measure the explained variance of benefits perception and harms perception by age, trust in institutions, and concern for health emergencies. The results shows that age, trust in institution, and concern for the COVID-19 emergency explain more variance of perceived benefits than of perceived harms of COVID-19 immunity certificates but the opposite regarding political orientation which explains perceived harms better than perceived benefits. The need for policy improvements is pressing because a large share of the world’s population remains unvaccinated. Moreover, our results can serve as vital information for similar health crises that may occur in the future. In addition, our results are expected to offer useful insights into public feelings around the use of digital health information tools.

## 1. Introduction

The global COVID-19 pandemic hit the world in early 2020. This led to worldwide restrictions on social life and freedom of traveling, while also damaging the global economy. In response to the widespread impact of the COVID-19 pandemic, countries across the world have proposed and implemented health certification policies (the so-called COVID-19 immunity certificates) that allow waivers on several restrictions (e.g., ability to travel, ability to access social venues, etc.) based on individuals’ infection/vaccination status or potential immunity. The purpose of such certifications was twofold: to restrict the access to social venues and high-risk situations (such as going to a restaurant or boarding an airplane) only to those individuals with a likely immunity to the virus and to promote the vaccination as well by giving an incentive to immunized persons. Most discussions around immunity- or infection-based documentation policies have focused on scientific plausibility, economic benefit, and challenges relating to ethics and equity. As COVID-19 vaccines are rolled out, attention has turned to confirmation of immunity and how documentation such as immunity passports can be implemented. However, the socio-cultural variability interacting with the implementation of COVID-19-related policies may hinder a one-size-fits-all approach [1]. Social science literature on policy implementation, discussing how health policies and guidelines unfold and are carried out in practice, highlights that this is not a straightforward undertaking that happens the same way everywhere for everyone [2,3]. Indeed, a relevant consideration for governments in deciding how to intervene to change people’s behavior is the attitude of the public towards such interventions and the extent to which any interventions are likely to be deemed as acceptable. This matters not only because levels of acceptability may critically affect the effectiveness of the health policy measure but also because accountable governments need to be aware of public attitudes if they want to act in a way that takes into consideration the population’s doubts and feelings. Indeed, the public following or adhering to health policies or guidelines does not imply an automatic process of acceptance; on the contrary, according to other studies in this field, people’s adoption of public health measures is intertwined with individuals’ characteristics—such as socio-demographics [4,5,6,7,8,9,10,11], COVID-19-related health status [6,8], attitudes towards vaccination [12], political orientation [4,6,13], health risk perception [7], and trust in institutions [8,14]—and attitudes towards such mandates in terms of potential perceived benefits or harms that they may provide them.

Previous studies on the public perception of COVID-19 immunity certifications [4,9,14,15,16] showed that reasons in favor of this measure were related to some individual and collective perceived benefits they could provide (i.e., infection reduction, vulnerable people’s health protection, safer access to public venues, public health safeguarding, and economy protection) [4,8,13,15,16]. On the other hand, research also showed some perceived harms and concerns related to this measure (i.e., privacy and freedom violation and inequality increase) [4,9,15,16]. 

Based on these premises, this study aimed at investigating the citizens’ reasons for accepting or rejecting the use of COVID-19 immunity certificates and at identifying the psychosocial factors that most predict the positive/negative individuals’ perceptions of this measure. 

## 2. Materials and Methods

### 2.1. Sample and Procedure

One thousand twenty-two Italian adults aged between 20 and 72 years old were recruited by a professional panel provider (Norstat Italia srl) by employing a stratified sampling strategy controlled for gender, age, geographical area of residence, size of the urban centre of the residence, employment, and wage. After providing their informed consent, the participants were asked to fill out an online survey (using a CAWI methodology); this study is part of a broader project (“Italian Citizens’ Food Habits Monitoring from a Consumer Psychology Perspective) aimed at monitoring Italian citizens’ habits. The study was designed as a cross-sectional survey, and data were collected between 29 January 2022 and 3 February 2022. At the time of data collection, in Italy, it was mandatory to show COVID-19 immunity certificates to join almost all social activities, including going to work, going to a bar or a restaurant, traveling by public transportation, and many others. 

### 2.2. Measures 

The participants were asked questions regarding:Socio-demographic questions such as gender, age, monthly family wage, and level of education. Education was recorded in 3 categories (1 = middle school or lower, 2 = secondary school, and 3 = degree or more);One question about whether they have ever contracted COVID-19 or not (proved by test);One question regarding their political orientation on a scale from 1 (far left-wing) to 10 (far right-wing) plus the option “I don’t take a position, I don’t care”. Answers were then recoded it into 4 categories (“left-wing”, “right-wing”, “center”, and no “political orientation”);One question assessing their level of trust in institutions on a 5-step Likert-type scale, where higher numbers corresponded to higher levels of trust;A question assessing their level of concern about the health emergency by asking them to indicate how concerned they are about the emergency from COVID-19 on a scale of 1 to 10;One question assessing their level of confidence about the effectiveness of vaccines in preventing infectious diseases;Eleven questions regarding their opinions on the COVID-19 immunity certificate policy measure, which stressed both potential benefits (7 items) and harms of this certification (4 items). Opinions were assessed on a 5-step Likert-type scale, where higher numbers corresponded to higher levels of agreement. Items were assigned to benefits or to harms according to whether they investigate the limitations or the advantages in a similar way to Lewandowsky [8].

The whole survey administered to participants is reported in Appendix A. 

### 2.3. Analyses

One index of perceived benefits and one index of perceived harms of COVID-19 immunity certificates was made by calculating the mean of the answers to the relative items. 

Then, Cronbach’s Alpha was calculated to test the internal consistency of the two indices, and we considered a good value if >0.60 [17].

Univariate outlier detection was then carried out with the MAD function [18,19], checking the z-scores, and displaying the data with boxplots (available in the Appendix B).

After testing the assumptions of normality with the Shapiro–Wilk test and the QQ-plot and the homoscedasticity with Levene’s Test, eight Welch’s ANOVAs [20] were then carried out to compare the perception of benefits and the perception of harms amongst males and females, those who had COVID-19 and those who had not, three groups with different education levels, and four groups with different political orientation. Then, Games–Howell post-hoc tests were run where relevant. The Eta-Squared index (η^2^) was calculated for each of the Welch’s ANOVAs to measure the effect size, considering values as recommended by Cohen [21].

Finally, two multiple linear regressions with the entering method were carried out to measure the explained variance of benefits perception and harms perception by the age, the trust in the institution, the concern for health emergencies, and the confidence level in effectiveness of vaccines to prevent infective disease. Residuals vs. fitted and residuals vs. leverage plots were made for diagnostic analysis and can be viewed in the Appendix C.

All the analyses were carried out using R in the R Studio environment; the script can be found in the attached material. Some minor variables required recodification and labelling with SPSS software (IBM SPSS Statistics 27.0.1.0, Armonk, New York, NY USA).

## 3. Results

### 3.1. Descriptive Statistics 

Overall, the sample included 1022 Italian citizens (50.7% female) aged between 18 and 70 years old (mean 46.9 with a standard deviation of 13.8). Table 1 shows the socio-demographic characteristics of the whole sample.

Cronbach’s α for the seven items related to the potential benefits of COVID-19 immunity certificates was 0.956, while for the four items related to the potential harms of COVID-19 immunity certificates was 0.902; items and their descriptive statistics are displayed in Table 2. No outliers were detected with z-scores; a few were found by displaying the data with boxplots—available in Appendix B—in the case of the variable “confidence level in vaccine efficacy”, and also a few in the case of the variable “concern for the COVID-19 emergency” were found using the MAD function [18,19]. Since none of these outliers were consistent in more than one of these three methods, we decided to not remove any of them.

### 3.2. Difference between Groups

All diagnostic plots show no issues with the regression models, so we have proceeded to the results’ production and interpretation.

Independence of the observations was guaranteed by the panel provider, while the homoscedasticity was violated just in two cases out of eight, i.e., benefits perception of COVID-19 immunity certificates on political groups and harms perception of COVID-19 immunity certificates on gender, and for this reason, Welch’s test was chosen instead of the standard ANOVA test. On the other hand, the Shapiro–Wilk test and the QQ-plots show that normality was violated in each of the eight cases; nevertheless, we considered this violation as not a serious threat to the reliability of our results because of ANOVA’s robustness to the violation of this assumption [22,23,24] and in light of a comparison showing no difference in the significance of the p-value with its nonparametric counterpart, i.e., the Kruskal–Wallis test, which, however, would have caused us to suffer a loss of the ability to interpret the results (results of this test can be found in the Appendix B).

Welch’s ANOVAs show a significant main effect of the political orientation both on benefits perceptions with a small effect size [F_(3, 493)_ =14.47; *p* < 0.001; η^2^ = 0.04] and on harms perceptions with a medium effect size [F_(3, 497)_ = 26.94; *p* < 0.001; η^2^ = 0.07]. In particular, post hoc comparisons show that the harms perception of COVID-19 immunity certificates is significantly lower (*p* < 0.001) for left-oriented people (M = 2.12, SD = 1.18) and higher (*p* < 0.001) for right-oriented people (M = 3.17, SD = 1.26); vice versa, the benefits perception of COVID-19 immunity certificates is higher (*p* < 0.001) for left-wing people (M = 3.79, SD = 1.05) and lower for right-wing people (M = 3.19, SD = 1.23).

In addition, Welch’s ANOVAs show a significant main effect of having contracted COVID-19 both on benefits perceptions with a small effect size [F_(1, 341.66)_ = 29.42; *p* < 0.001; η^2^ = 0.28] and on harms perceptions [F_(1, 351)_ = 30.23; *p* < 0.001, η^2^ = 0.27]. The results show a statistically significant difference with *p* < 0.05, with higher benefits perceptions of COVID-19 immunity certificates for those who have contracted COVID-19 (M = 3.52, SD = 1.17) and lower for those who have not contracted it (M = 3.03, SD = 1.17); on the other hand, concerning the harms perception of COVID-19 immunity certificates, the results show a statistically significant difference with *p* < 0.05, with a lower rate for those who have contracted COVID-19 (M = 2.59, SD = 1.28) and higher for those who have not contracted it (M = 3.11, SD = 1.21).

Finally, results show no significant effects of the education on benefits perceptions (*p* = 0.081) nor harms perceptions (*p* = 0.166) and no significant effects of the gender on benefits perceptions (*p* = 0.361) nor harms perceptions (*p* = 0.296).

### 3.3. Multiple Linear Regression Models

The multiple linear regression with age, trust in institution, concern for the sanitary emergency, and confidence level for efficiency of vaccines to prevent infective disease as independent variables and perceived benefits as the dependent variable returned a significative model that explains benefits perceptions [F_(4, 1017)_ = 313.00, *p* < 0.001] with an *R^2^* of 0.552. All the independent variables had a significative effect on the dependent variable (*p* < 0.001) and predicted that a benefits perception of the COVID-19 immunity certificate was equal to 2.74 + 0.55 (confidence level in vaccine efficiency to prevent infective disease) 0.37 (trust in institution) + 0.12 (age) + 0.10 (concern for the COVID-19 emergency).

Then, the multiple linear regression with age, trust in institution, concern for the sanitary emergency, and confidence level for efficiency of vaccines to prevent infective disease as independent variables and perceived harms as the dependent variable returned a significative model that explains harms perceptions, but with a lower amount of variance explained [F_(4, 1017)_ = 97.23, *p* < 0.001] with an *R*^2^ of 0.277. Neither trust in institution (*p* = 0.25) nor concern for the COVID-19 emergency (*p* = 0.06) have a significant effect on the dependent variable harms perception, while age and confidence level for efficiency of vaccines to prevent infective disease do have a significant effect (*p* < 0.001). The predicted harms perception of the COVID-19 immunity certificate was equal to 2.88–0.61 (confidence level for efficiency of vaccines to prevent infective disease) and 0.16 (age).

## 4. Discussion

In this study, we indicate details that could be useful to develop better communication strategies to engage people to better accept restrictive measures for the sake of a common good. Those details, hence, can be considered to envision specific targets to propose-tailored communications that are more relevant and therefore more effective [25].

In fact, considering perceived benefits separately from perceived harms has shown how there are variables that influence one more than the other. Firstly, results show that considering the multiple linear regression model with age, confidence level for efficiency of vaccines to prevent infective disease, trust in institution, and concern for the COVID-19 emergency, the latter two significatively explain the variance in the perceived benefits while not in the perceived harms of COVID-19 immunity certificates, but the opposite regarding political orientation, which explains perceived harms better than perceived benefits.

However, these results show that an initial classification of targets by their demographic variables such as gender or education is not too effective in skimming for perceptions of harms and benefits. In fact, neither gender nor education turn out to have a significant effect in perceptions of benefits or harms. This result is apparently in contrast with previous studies [9,10,26], but while those studies investigated opinions regarding a hypothetical immunity certificate, in our study, such a system was already valid at the time of the survey. 

However, the results show age as a significant sociodemographic explanatory variable for benefits perceptions, which increases with age. This is consistent with previous studies that show a better acceptance for older people [5,10] and could be a relevant result to set up more communication strategies specific for younger people, for example, using infographic messages [27].

Then, it might be expected that having had the disease—and having experienced firsthand the risk and the discomfort—should increase the perceived benefits and reduce the perceived potential harms of the COVID-19 immunity certificate that, in fact, is a measure to solve the emergency, but looking at the results, it is clear that those who have contracted COVID-19 perceive more of the harms and less of the benefits compared to those who have not. Indeed, this result is in line with a previous study and could be due to the fear of being isolated for a period of time because of this certification system, especially from those who are most likely to be temporarily immune [28]; in any case, it could be worthwhile to better investigate the reason for this with further studies.

Instead, it seems more intuitive that perceived benefits increase as concern about the health emergency increases, but interestingly, it is not as effective at influencing perceptions of harms. This, since the percentage of the population concerned changes over time depending on events [29], may imply that during the most worrisome periods it is easier to introduce restrictive measures.

Regarding political orientation, it turns out that right-wing people perceive more of the harms and less of the benefits of the green pass compared to left-wing people. In another study, it was found that more conservatives scientists would accept the introduction of an immunity passport better [4]; this difference could be addressed to the special kind of population and the fact that people of right-wing orientation could not be the same [30]. Although predictable, trust in institutions turns out to be a good predictor of the perceived benefits of the green pass and, therefore, of a restrictive measure for the common good, which is why it may be important to monitor an index of trust toward institutions, such as with a social media analysis [14], and improve this condition [31] to increase acceptance of policies for the good of the community [8,32]. However, it is interesting that the perceived harms of such a measure are not addressed, so it might be interesting to investigate the profile of those who perceive more harms than benefits in more depth. Finally, the results of the model clearly show the role of the confidence level for efficiency of vaccines to prevent infectious disease; this, in addition to finding confirmation with very recent studies also in the Italian context [33], brings to attention the importance of continuing in the effort to reduce vaccine hesitancy by listening to their doubt and by creating space for dialogues with the citizens that can provoke their engagement with the health and social system. The need for policy improvements is pressing because a large share of the world’s population remains unvaccinated, and recent scientific evidence showed a waning of the immune response over time. Moreover, even beyond the current pandemic, our results can serve as vital information for similar health crises that may occur in the future. In addition, our results are expected to offer useful insights into public feelings around the use of digital health information tools. 

## 5. Limitations

This study has some limitations that should be considered. The data presented in this study are self-reported and are partly dependent on the participants’ honesty and recall ability; thus, they might be prone to recall, declaration, or desirability biases. Secondly, the sample was limited to the Italian context, and therefore, the collected responses might not be generalizable to other countries. Future research should consider the results of this study with different populations. 

## Figures and Tables

**Table 1 vaccines-10-01501-t001:** Sample characteristics.

	%	n
**Gender**		
Male	49.3%	504
Female	50.7%	518
**Employment**		
Entrepreneur/freelancer	12.4%	127
Manager/official	3.8%	39
Employee/military/teacher	22.1%	226
Worker/shop assistant/apprentice	18.1%	181
Householder	15.0%	153
Student	5.3%	54
Retired	7.9%	81
Unoccupied	15.4%	157
Other	0.3%	4
**Education**		
Middle school or lower	17.4%	178
Secondary education	56.1%	574
Degree or more	26.4%	270
**Geographical Area**		
Northwest	26.3%	269
Northeast	18.6%	190
Center	19.7%	201
South	35.4%	362
**Living center’s size**		
Up to 10.000	32.1%	328
Between 10.001 and 100.000	44.0%	450
Between 100.001 and 500.000	10.9%	111
Above 500.001	13%	133

**Table 2 vaccines-10-01501-t002:** Survey items assessing perceived harms and benefits of the immunity certification implementation.

Item	Mean	S.D.	Asymmetry	Skewness
**Benefits Perception (α = 0.952)**	**3.42**	**1.18**	**−0.72**	**−0.45**
I think COVID−19 immunity certificate is an effective measure to reduce infections	3.28	1.35	−0.45	−0.96
I think COVID-19 immunity certificate is important to protect the health of the most fragile people	3.59	1.32	−0.70	−0.59
Since the existence of the COVID-19 immunity certificate I feel safer going to public places.	3.19	1.25	−0.39	−0.74
I think COVID-19 immunity certificate is important to protect public health.	3.52	1.31	−0.65	−0.62
I think it’s fair to prevent people who don’t have COVID-19 immunity certificate from entering workplaces.	3.46	1.40	−0.57	−0.92
I think COVID-19 immunity certificate is important to protect the economy.	3.27	1.31	−0.37	−0.88
I think it is fair to prevent access to recreational and social gathering places such as restaurants, nightclubs, and stadiums to those who do not have COVID-19 immunity certificate	3.61	1.35	−0.74	−0.61
**Item**	**Mean**	**S.D.**	**Asymmetry**	**Skewness**
**Harms Perception (α = 0.902)**	**2.70**	**1.28**	**−0.27**	**−1.06**
I think it is fair to prevent access to recreational and social gathering places such as restaurants, Nightclubs, and stadiums to those who do not have COVID-19 immunity certificate	2.54	1.47	0.42	−1.22
COVID-19 immunity certificate is a violation of citizens’ privacy	2.72	1.42	0.20	−1.24
I think the COVID-19 immunity certificate is a way for the government to control the citizens	2.67	1.49	0.30	−1.30
COVID-19 immunity certificate strongly violates personal freedom	2.87	1.45	0.09	−1.31

## Data Availability

The dataset used for the analysis in this paper is available by emailing the corresponding author (michele.paleologo@unicatt.it). The R-script can be found in the attached materials.

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
