# Peer review of "Public Perceptions of Harms and Benefit of COVID-19 Immunity Certificate: A Cross-Sectional Study in the Italian Setting"

_vaccines, 2022, doi:10.3390/vaccines10091501_

Round 1

Reviewer 1 Report

I read the paper by Barello and colleagues with great pleasure.

The paper is certainly interesting, worthy of publication in Vaccines and well written. My compliments.

I have a few small questions to ask, which I would be pleased if the authors could clarify:

1) when it comes to political orientation, I have seen that they use a scale on a base of 10 from left to right. What I am wondering, knowing as an Italian how Italians experience politics, is whether one could not ask for voting orientation (i.e. voting Lega or PD). Wouldn't that have been a more accurate indicator in your opinion?

2) don't you think that a generic question about trust in previous vaccinations (e.g. flu and/or how much they believe in the mandatory vaccination and/or the Lorenzin law) and/or how much trust they have in alternative therapies (i.e. naturopathy and the like) could have enriched the model and given additional information? Can you discuss this?

Finally, I am begging you pardon and I recognise that I am punctilious, but throughout the paper I have seen 'Covid-19' and 'COVID-19' written indifferently. I would ask you to bring everything into line with this second spelling. Also check the paper for some typos.

Author Response

Response to Reviewer 1 Comments

Point 1: when it comes to political orientation, I have seen that they use a scale on a base of 10 from left to right. What I am wondering, knowing as an Italian how Italians experience politics, is whether one could not ask for voting orientation (i.e. voting Lega or PD). Wouldn't that have been a more accurate indicator in your opinion?

Response to point 1: Thank you for your comment, we decided not to ask for voting intentions mainly for two reasons. The first of which is the risk of receiving many NA and wrong answers due to the sensitivity of the question with high social desirability. The second, perhaps more important, concerns the high percentage of abstentionism in the Italian context which means that there are many citizens who can more easily identify with the right and the left but less with a particular party.

Point 2: Don't you think that a generic question about trust in previous vaccinations (e.g. flu and/or how much they believe in the mandatory vaccination and/or the Lorenzin law) and/or how much trust they have in alternative therapies (i.e. naturopathy and the like) could have enriched the model and given additional information? Can you discuss this?

Response to point 2: Thank you for your comment, we didn’t consider in our hypotesis also because a lack of literature supporting this relationship at the time of the survey development. But, since you raised the issue, we agreed that it could be a really significant variable to include in our model, and so it was. Indeed, given that in the original survey there was also a question regarding the level of confidence of vaccine efficiency to prevent infective disease we included in the analysis. Results and discussion about it can be now found in the revised version of the paper.

Point 3: Finally, I am begging you pardon and I recognise that I am punctilious, but throughout the paper I have seen 'Covid-19' and 'COVID-19' written indifferently. I would ask you to bring everything into line with this second spelling. Also check the paper for some typos.

Response to point 3: Thank you for your comment, since details create the big picture, we’ve aligned everything with the second spelling as you suggested and we checked at our best for the presence of typos.

Reviewer 2 Report

1. The authors should mention the behavior of data based on quantile-quantile, box, and kernal plots.

2. Did the data contain extreme or outlier observation?. Explain. 

3. To apply ANOVA test, the authors must test the normality and the homogeneity properties. Did the authors do this?. Explain. 

4. A cross-sectional survey between 29 January 2022 and 03 February 2022 -----. Did this period is enough for this study?. I think it is a short period. 

5. Regarding the regression model, the authors should sketch the residual plot. 

6. The authors should plot the scattering of this data "Spread out" to determined the shape of data.

7. The trusted value is 0.101. It is a small value. The authors should use another regression model. 

Author Response

Response to Reviewer 2 Comments

Point 1: The authors should mention the behavior of data based on quantile-quantile, box, and kernal plots.

Response to point 1: Thank you for your comment, We have produced the we generated new graphs according to your suggestions and included in the attached material. We have then mentioned the behavior of the variables using these graphs where we felt it was necessary.  

Point 2: Did the data contain extreme or outlier observation?. Explain

Response to point 2: Thank you for your comment, we performed univariate outlier detection both by the graphical display and by checking z-scores. As also suggested by your first comment, we made box plots for each variable included in the analyses, and only in the case of the variable "degree of agreement with the effectiveness of vaccines in general for preventing infectious diseases." - just introduced on the advice of referee No. 1 - we can observe the presence of an outlier. However, it can be inferred from the z-scores that the minimum score equals 2.40 very far from the commonly used threshold value of 3 and also less than the value of 2.58 within which in the case of a normal distribution 95% of cases fall. Then, for further confidence, we used the MAD function for outlier detection and were detected 98 outliers in case of the variable concern for COVID-19 emergency . Since none of these outliers were consistent with more than one of these three methods we decided to not remove any of them.

Point 3: To apply ANOVA test, the authors must test the normality and the homogeneity properties. Did the authors do this?. Explain

Response to point 3: Thank you for your comment, we apologize for not having clarified these key issues. Yes, we tested the assumptions: the independence of the observations is ensured by the provider who administered the online questionnaire to participants through which data were collected.

Then, regarding homoscedasticity, this is violated in two out of eight cases (benefits perception on gender and benefits perception on political groups) for this reason it was still decided to conduct welch's test in all cases as suggested by Delacre and colleagues (2019). Instead, checking with the Shapiro-Wilk test, the assumption of normality is violated in all eight cases. From the QQ-plots, anomalies can be seen in the heads and tails of the graph. Nevertheless, in light of the literature in support of the ANOVA's robustness to the violation of this assumption (Blanca et al., 2017; Knief & Forstmeier, 2021; Schmider et al., 2010) and in light of a comparison showing no difference in the significance of the p-value with its nonparametric counterpart, i.e., the Kruskal-Wallis test (which, however, would have caused us to suffer a loss of ability to interpret the results). As mentioned above in the supplementary material, the R script from which the comparison between the Welch test and the Kruskall-Wallis test can be performed will also be attached.

Point 4: A cross-sectional survey between 29 January 2022 and 03 February 2022 -----. Did this period is enough for this study?. I think it is a short period. 

Response to point 4: Thank you for your comment. However,this is not a longitudinal study, and we have therefore tried, through the provider who administered the questionnaires, to reduce as much as possible the distance between the beginning and the end of the administration period precisely to reduce any variability due to changes and/or events related to the issue at hand.

Point 5: Regarding the regression model, the authors should sketch the residual plot. 

Response to point 5: Thank you for your advice, we added the residuals vs fitted the residuals vs leverage plots to appendix and mentioned it in the methods and results section. 

Point 6: The authors should plot the scattering of this data "Spread out" to determined the shape of data.

Response to point 6: Thank you for your comment, we added all this plots in the appendix C and men.

Point 7: The trusted value is 0.101. It is a small value. The authors should use another regression model. 

Response to point 7: Since the other referee asked us to consider the possible impact of a question concerning the level of confidence in vaccination in general and/or the inclination towards other forms of medicine such as naturopathy, and since in the administered questionnaire was present the question "am I confident in the efficacy of vaccines in preventing infectious diseases" (5-steps likert scale), we decided to add it to our two multiple linear regression models. Because of this, both R2 went up and the one you understandably questioned went from 0.101 to 0.277.

Round 2

Reviewer 2 Report

More efforts have been done in the revised version. All comments have been discussed in detail. Thus, I accept publication this paper in the current form.